META-RESEARCH

# Reader engagement with medical content on Wikipedia

**Abstract** Articles on Wikipedia about health and medicine are maintained by WikiProject Medicine (WPM), and are widely used by health professionals, students and others. We have compared these articles, and reader engagement with them, to other articles on Wikipedia. We found that WPM articles are longer, possess a greater density of external links, and are visited more often than other articles on Wikipedia. Readers of WPM articles are more likely to hover over and view footnotes than other readers, but are less likely to visit the hyperlinked sources in these footnotes. Our findings suggest that WPM readers appear to use links to external sources to verify and authorize Wikipedia content, rather than to examine the sources themselves.

**LAUREN A MAGGIO\*, RYAN M STEINBERG, TIZIANO PICCARDI AND JOHN M WILLINSKY**

**\*For correspondence:** lauren. maggio@usuhs.edu

**Competing interests:** The authors declare that no competing interests exist.

## Introduction

Wikipedia is a freely available online encyclopedia that intends to provide "every single person on the planet free access to the sum of all human knowledge" (**Wikimedia Foundation, 2004**). To meet this mission, thousands of volunteer editors have created almost six million English-language Wikipedia pages. Many of these pages include hyperlinked footnotes to the sources used to assemble and verify the content (**Redi et al., 2018**; **Wikipedia Foundation, 2019a**). These linked references do not only serve as sources of authority for Wikipedia content (**Fallis, 2008**), but also offer readers a gateway to further learning, with this opportunity enhanced by the growing degree of public access to research literature (**Piwowar et al., 2018**).

Wikipedia is proving to be a leading source of health information (**Heilman and West, 2015**; **Laurent and Vickers, 2009**), and reader engagement with a page's references can provide opportunities to understand a diagnosis or inform a conversation with their physician. Wikipedia's health-focused pages, which are maintained by WikiProject Medicine (WPM) editors, are thought to meet a high standard of quality and rigor (**James, 2016**; **Maskalyk, 2014**; **Trevena, 2011**). For example, WPM provides a

series of resources for its editors, including "guidelines and policies" for reliable sources, and advice on how to avoid conflicts of interest. In its advice for reliable sources, WPM recommends using "review articles (especially systematic reviews) published in reputable medical journals" (**Wikipedia Foundation, 2019b**). Wikipedia then provides tools to standardize the bibliographic representation and linking to external sources.

These external sources are particularly relevant for students and practitioners in medicine and other health-related professions, who are active Wikipedia readers (**Rössler et al., 2015**; **Scaffidi et al., 2017**; **Back et al., 2016**; **Egle et al., 2015**; **Allahwala et al., 2013**). These readers are familiar with the research literature, and expected to engage in evidence-based medicine (**Guyatt et al., 2015**). To this end, faculty at several health professions schools, such as schools of medicine and pharmacy, teach courses on editing Wikipedia (**Azzam et al., 2017**; **Joshi et al., 2019**; **Apollonio et al., 2018**).

In a previous study, we investigated Wikipedia as a gateway to further reading through external links by tracking referrals from Wikipedia to research article DOIs (Digital Object Identifier) through Crossref, combined with

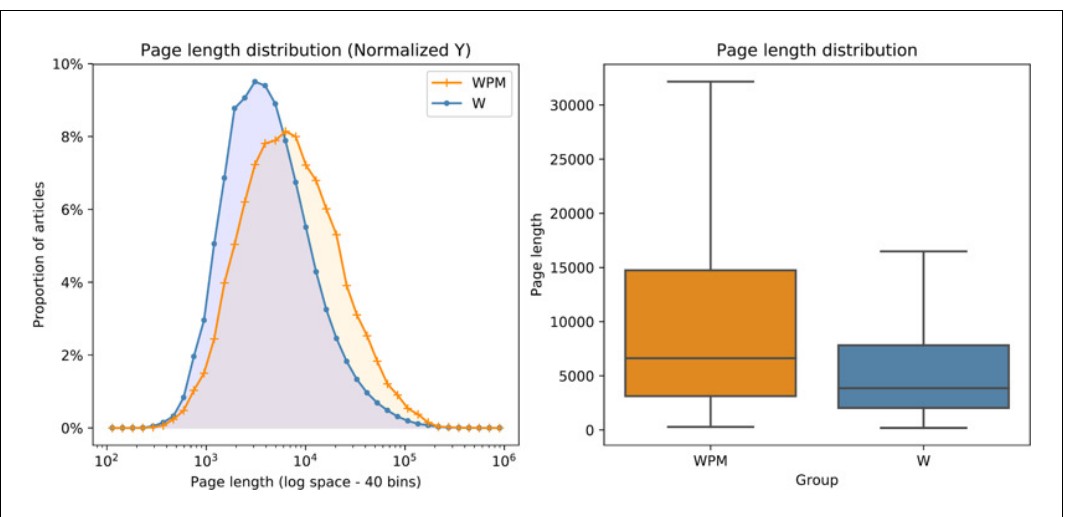

**Figure 1.** Page length of WikiProject Medicine (WPM) and other Wikipedia (W) pages. Distribution of page lengths in characters for WikiProject Medicine (WPM) pages and the rest of Wikipedia (W) on April 20th, 2019. The difference between the two distributions is statistically significant according to Mann–Whitney U test (p<0.001, two-tailed).

pageview information from Wikimedia (*Maggio et al., 2017*). However, with the data available at the time, we could not determine if a click to an external reference with a DOI originated from a WPM page.

To learn more about reader engagement with Wikipedia's external references, this study leverages Wikimedia's newly created infrastructure for data collection to compare engagement with external links by readers of WPM pages with that of readers of the rest of Wikipedia. The study is guided by two research questions:

**RQ 1**: To what extent do WPM pages differ on average from the rest of Wikipedia pages?
**RQ 2**: To what extent do the behaviors of WPM readers with external links differ on average from readers' behaviors with the rest of Wikipedia?

## Methods

With the approval and support of the Wikimedia Foundation (WMF), we collected the data presented in this study between March 22nd – April 22nd, 2019 from Wikimedia's Event Logging system and the production MediaWiki database. The data remained within that system, as required by WMF, for a year before being deleted (see Acknowledgements). We anonymized the aggregated data by removing IP addresses, any identifying browser information, and reader-sessions associated with page edits. While this data is not publicly available, data

may be available upon request from the WMF research team. To gain access to the data, researchers should review the WMF Research team's current procedures for data requests (https://www.mediawiki.org/wiki/wikimedia_research/research_and_data). The WMF Research Team will evaluate the request based on Wikimedia's data access criteria. The code utilized to collect and analyze the data, however, is organized and made publicly available in a collated series of Jupyter notebooks in GitHub (*Steinberg and Picardi, 2019*; copy archived here).

### Data collection

The data is drawn from English Wikipedia pages in the main namespace, a designation that contains the encyclopedia proper. Wikipedia pages or topics were identified as being part of two main groups: WikiProject Medicine pages (WPM) and the rest of Wikipedia (W). To be included in the study, the pages from each group had to have at least one external link in the externallinks table. The categorylinks table was used to define the WPM pages, with each possessing a Talk page bearing the category "All WikiProject Medicine articles." Both the externallinks and categorylinks tables were queried twice (April 1st, 2019 and April 20th, 2019) during a 32 day study period (March 22nd – April 22nd, 2019).

For determining the number of pages, length of pages, the number of external links, and the

number of "freely accessible" links added by editors as sources, a single day's worth of database and XML dump files were captured from late in the study period (April 20th, 2019). As the database and XML dump files had only 0.5% more external links than on April 1st, 2019, the sample from April 20th was felt to be sufficiently representative to serve as the source for all static data counts. The external link count, which is based on MediaWiki's externallinks table, does not include interwiki links, representing abbreviated forms of commonly-used internal and external links, which limits the accuracy of external link counts for both WPM and W. The event logging system this study relied on similarly omitted data from interwiki links, meaning the definition of an external link used across this study is consistent.

Pageview data was gathered from the wmf. pageview_hourly table. WMF employs methods to identify bot traffic in pageview data, which was excluded in our analysis. In reporting the data collected over a 32 day period (March 22nd – April 22nd, 2019), the raw counts were divided by 32 to create a count approximating a "daily average" for these counts, in light of this serving as a common measure of internet traffic.

Reader engagement with external links was gathered from Wikimedia's Event Logging system using the CitationUsage schema, instrumented by Wikimedia's programmers, following a month of piloting and refinement for this study. The CitationUsage schema collected all sessions with reader engagement, except for those involving anonymous Wikipedia editors (21 sessions out of 72,953,065 total, which translated into the removal of 34 citation events out of a total of 113,520,376). For the entire study period (March 22nd – April 22nd, 2019), the CitationUsage schema captured the following types of engagement: (a) clicking an external link; (b) clicking on a reference link listed among a page's set of references; (c) clicking a footnote link leading to a reference on the page; (d) hovering over a reference link (defined as a reader's cursor lingering over a link for 1000 milliseconds or more, revealing a rollover label); (e) time from pageview to event (Table 1). Additionally, the CitationUsage schema captured the clicking of links bearing the "freely accessible" icon.

### Analysis

Descriptive statistics were calculated using Excel (Redmond, WA). Inferential statistics were calculated to determine the significance in the difference between engagements in WPM and W using the Python library SciPy. In our analysis, we verified the statistical differences between WPM and W page for the following parameters: (a) time to first engagement with a link, (b) length of the articles, and (c) the number of page loads per event. For parameters (a) and (b) we used a normality test to determine the type of statistical test to apply. In both comparisons, after rejecting the normality hypothesis, we applied Mann-Whitney U-Test, a non-parametric test that does not require normality assumptions on the two distributions. For the third parameter (c), we compared the two groups over all possible combinations of access method and type of engagement with a two-tailed Fisher's exact test. To explore and visualize the data, we used respectively Spark/Pandas and the library Seaborn. The complete analysis is available in the project's publicly accessible Github repository (*Steinberg and Picardi, 2019*).

### Stakeholder engagement

This study was possible through collaboration with the WMF research team beginning in 2018, with our publishing of a research proposal to Wikimedia's Meta-Wiki (*Maggio et al., 2018*). Members of the Wikipedia community, including members of WPM, were invited to provide feedback and pose questions. Over several months, we held teleconferences and in-person meetings with the WMF research team to understand

**Table 1.** Types of engagement with external links

Different events captured by the CitationUsage Schema that reflect how readers of Wikipedia pages engage with external links.

| Engagement type | General description |
| --- | --- |
| External click | A click on a link located on a Wikipedia page leading to a web page outside of Wikipedia. |
| Hover | An event that occurs when a reader hovers over a link for at least 1000 milliseconds on a Wikipedia page. |
| Footnote click | A click of an internal footnote link – [1] – that takes the reader to the reference section at the bottom of the Wikipedia page. |
| Up click | A click of an arrow – ˆ – that takes the reader from the reference at the bottom of the Wikipedia page back to the citation in the main text. |

**Table 2.** Factors that distinguish WPM pages from the rest of Wikipedia.
Differences between WPM pages and pages in the rest of Wikipedia (W) based on data collected on April 20th, 2019.

|  | WPM | W |
| --- | --- | --- |
| Wikipedia pages | 34,324 | 5,839,083 |
| Pages with external links | 32,609 | 5,210,746 |
| External links | 945,645 | 60,851,396 |
| Links per page (with links) | 29.0 | 11.7 |
| Page length (characters) | 13,084.9 | 7,676.4 |
| Characters per link (pages with links) | 450.3 | 657.3 |

Wikimedia's infrastructure, parameters of data use (e.g., best practices for accessing the data securely; duration of data access), and the expectations of the Wikipedia community. Prior to data collection, a WMF research team member posted to Wikipedia's Village Pump, a set of Wikipedia pages dedicated to discussing technical issues and policies, an announcement of the data collection (*Redi, 2019*). The announcement invited the Wikipedia community to post public comments and provided contact information for expressing concerns about the research.

## Results

### *Wikipedia pages and external links*
This study compares readers' engagement with the pages curated by WikiProject Medicine (WPM) to their engagement with the rest of the English edition of Wikipedia (W). At the time of this study, WPM represented 34,324 pages (i.e., subject or topic entries), while the rest of W had 5,839,083 pages (*Table 2*). WPM pages possessed more than twice as many links to external references and other sites than the rest of Wikipedia. WPM pages were 13,084.9 characters in length on average (SD = 19,3780.4; median = 6,6280.0; IQR = 11,640), which was 70.1% longer than the average page in the rest of Wikipedia at 7,676.4 characters

(SD = 13,6320.4; median = 3,8650.0; IQR = 5,789) (*Figure 1*; Table 6). WPM pages also demonstrated a greater "link density" with an external link appearing on pages with links every 450.3 characters, compared to a link every 657.3 characters for the rest of Wikipedia.

The biomedical literature, including research reviews and studies, was a leading source of WPM external links. For example, the most common hostname was "www.ncbi.nlm.nih.gov," accounting for 25.2% of the WPM's external links. This indicates that research papers found through the PubMed database run by the National Institutes of Health (NIH) are likely to be cited. It was followed by the hostnames "www.worldcat.org" with 2.7%, and "www.google.com" with 1.2% of the external links. As for the hostnames that prevailed among external links in the rest of Wikipedia, the three leading hostnames were "tools.wmflabs.org" (where Wikimedia hosts tools developed to assist editors) at 3.5%, "www.google.com" at 2.9% and "books.google.com" at 2.0%, while "www.ncbi.nlm.nih.gov" still accounted for 1.8% of the external links outside of WPM.

### *Pageviews and events*
Readers viewed 5,875,470.4 WPM pages a day on average, compared to 228,445,128.4 pages for readers of the rest of Wikipedia (*Table 3*). While the number of WPM pages is 0.6% of W

**Table 3.** Comparing pageviews between WikiProject Medicine (WPM) and other Wikipedia (W) pages.
The average number of pageviews WikiProject Medicine (WPM) and the rest of Wikipedia (W) received per day for different types of devices between March 22nd and April 22nd, 2019.

|  | WPM (%) | W (%) |
| --- | --- | --- |
| Pageviews on desktop device | 1,957,821.6 (33.3) | 97,956,273.1 (42.9) |
| Pageviews on mobile device | 3,917,648.8 (66.7) | 130,488,855.3 (57.1) |
| Total number of pageviews | 5,875,470.4 (100) | 228,445,128.4 (100) |

pages, the number of pageviews that WPM received from readers was 2.6% of that for W. This suggests that WPM pages were viewed more than four times as frequently as the rest of Wikipedia. The majority of these views for both WPM and W took place on mobile devices, with WPM readers the heavier users of mobile devices in accessing Wikipedia.

We also found that readers spent more time on a page before first clicking an external link after loading a WPM page. The WPM reader's median time of 47.5 s (SD = 1.2; mean = 82.5; IQR = 147,306) was 44.8% longer than the median time of 32.8 s (SD = 5.6; mean = 39.1; IQR = 86,424) for W readers (Table 6). The difference between the two distributions is statistically significant according to Mann–Whitney U test (p<0.001, two-tailed).

Among the four types of events recorded, WPM readers were more likely to hover over a footnote or other link, especially on their desktop devices, and more likely to click on footnote links, compared to W readers (*Table 4*). On the other hand, W readers were more likely to click on external links than WPM readers, favoring their mobile devices in that regard. The WPM readers are interested, it appears, in seeing what evidence is being drawn upon in making the statements set out on a WPM page rather than trying to view the source compared to readers of the rest of Wikipedia.

Among the external links in WPM, 22.1% of the research citations drawn from PubMed are labeled as "freely accessible", and on desktop devices include a link bearing an open access icon (a green open lock, with the rollover label "freely accessible"), as shown in *Figure 2*. However, there was no evidence to suggest that links with these icons were clicked more frequently than other links. This may be, in part, because the typical research citation has three or four links leading to (a) the article on the publisher's site (DOI link); (b) its PubMed entry (PMID link); and (c) PubMed Central (PMC link and article title link) if the article is open access.

To further compare event data (i.e. the type of engagement with external links) between WPM and W, the number of pageviews per event was calculated to determine how many pages readers viewed before engaging with an external link (*Table 5*). Overall, readers of WPM pages were more likely (56.6 pageviews per event) to engage with a link than W readers (66.5 pageviews per event). The differences in engagement were most pronounced with WPM readers hovering over a link on their desktop device. Yet, W readers more frequently clicked external links, with the difference especially notable when on their mobile device, where they were almost twice as likely to click an external link than WPM readers. This suggests that the external links in WPM may not be as mobile-friendly. The up click, in which a reader clicks the link from a footnote back (up) to the text, was not a function that appeared on the iOS and Android mobile devices we tested, as the footnote appears at the bottom of the screen when clicked and disappears on touching the text (*Figure 2*).

## Discussion

The Wikipedia pages maintained by WPM are longer, possess a greater density of external links (references), and are viewed considerably more often, on average, than the pages on the rest of Wikipedia (*Table 6*). The popularity of the pages also speaks to the valuable contribution that WPM makes to Wikipedia's role as a source of health and medical information, as well as the level of trust that WPM readers have in Wikipedia's coverage of health and medical topics.

**Table 4.** Frequency of different types of link engagement per day.
The average number of times per day that readers of WPM and W pages engaged with external links using one of the event types captured from March 22nd to April 22nd, 2019.

| Event type | WPM | | | W | | |
|---|---|---|---|---|---|---|
| | Total (%) | Desktop (%) | Mobile (%) | Total (%) | Desktop (%) | Mobile (%) |
| Hover over link | 48,748.9 (46.9) | 45,814.8 (60.3) | 2,934.1 (10.5) | 1,122,704.0 (32.7) | 1,057,982.0 (47.2) | 64,722.0 (5.4) |
| Footnote click | 27,739.4 (26.7) | 10,948.8 (14.4) | 16,790.6 (60.3) | 722,131.0 (21.0) | 235,245.0 (10.5) | 486,886.7 (40.7) |
| External click | 25,811.9 (24.9) | 17,792.3 (23.4) | 8,019.7 (28.8) | 1,557,125.0 (45.3) | 915,445.1 (40.9) | 641,676.4 (53.6) |
| Up click | 1,539.5 (1.5) | 1,422.8 (1.9) | 116.4 (0.4) | 34,738.0 (1.0) | 31,230.1 (1.4) | 3,508.1 (0.3) |
| All events | 103,839.7 (100) | 75,978.7 (100) | 27,860.8 (100) | 3,436,698.0 (100) | 2,239,902.2 (100) | 1,196,793.2 (100) |

In addition, the readers of WPM pages not only engage more often with external links, but spend longer on the page before engaging with the links, compared to readers of the rest of Wikipedia (*Table 6*). WPM readers are more likely to hover over references, causing them to appear in a window, and more likely to view the footnotes than other readers. But readers of WPM pages click on external links less often than readers of other parts of Wikipedia.

This behavior pattern suggests that readers are more interested in validating the external references. This interest may arise from readers who are relatively new to these pages and are in the process of reassuring themselves as to the scientific basis of the content they are consulting. But it may also arise from readers who are trained, or are in the process of being trained, in health professions where they are required to critically appraise and integrate health information into the care of their patients. These evidence-based practitioners are able to judge a lot about the quality of information from the bibliographic reference to the sources, which they view by hovering over or going to the footnote. In addition to being able to see at a glance (or a hover) whether the cited work is a review (which is recommend by Wikipedia as being the "ideal source"), readers are able to identify publication date, language, publication venue (i.e., journal name), and availability (via open access) (*Figure 3A*).

In addition, in some instances it is possible to obtain from a citation details on patient populations, medical interventions (e.g., treatments, diagnostic tests), and outcomes involved in the study or whether the source is a standard medical textbook in the field (*Figure 3B*). The greater attention that WPM readers are paying to citations could well be a function of their educational backgrounds, reflecting the demonstrated use of WPM by health professionals, students and practitioners. Although not the focus of this study, the educational value of internal links, in which terms on a WPM page are hyperlinked to other Wikipedia pages or to VideoWiki, should also be noted as a further strategy for maintaining the comprehensive informational and educational quality of WPM pages.

This still leaves the question of why WPM readers are less likely to click through to the external source than readers of the rest of Wikipedia. This question is particularly relevant given the interest demonstrated by WPM readers in viewing the citations (by hovering or clicking on footnote numbers), as well as by the studies (cited above) demonstrating that health professionals use WPM pages, which suggests a potential capacity for further learning through the external links. One possibility is that readers are deterred from clicking on external WPM links in light of our finding that only a fifth of the research cited in WPM pages are publicly accessible, with the rest placed by publishers behind a paywall for subscribers or credit card access. Earlier studies have found that it is common for readers, including physicians, who encounter a paywall to be reluctant to explore the literature further (*Moorhead et al., 2015*; *Maggio et al., 2016*). It is possible that if all the research

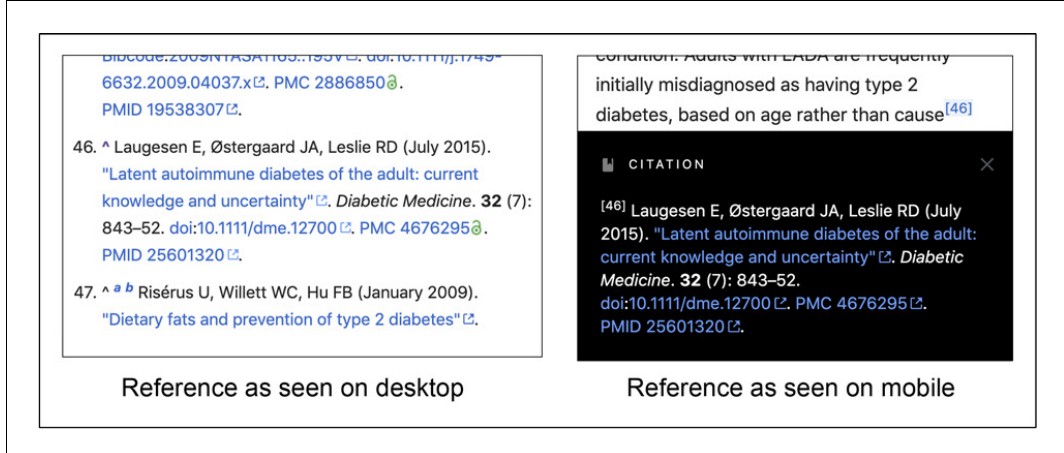

**Figure 2.** Open access reference displayed in Wikipedia. A research study cited on Wikipedia's "diabetes" page displayed on desktop device with "freely accessible" icon for PubMed Central (PMC) link and on a mobile device without the open access icon. There is an "up click"(^) in front of the citation on the desktop device, which returns reader back to footnote 46 in the text.

**Table 5.** Number of pageviews per engagement event.
The frequency of each event per day was divided by the average number of daily pageviews for WPM and W pages from March 22nd to April 22nd, 2019. The lower the number of pageviews per event the greater the event frequency. Difference between each pair of WPM and W distributions is statistically significant as derived from Fisher's exact test (p<0.001, two-tailed).

| Pageviews/event | WPM | | | W | | |
|---|---|---|---|---|---|---|
|  | Total | Desktop | Mobile | Total | Desktop | Mobile |
| Hover over link | 120.5 | 42.7 | 1,335.2 | 203.5 | 92.6 | 2,016.1 |
| Footnote click | 211.8 | 178.8 | 233.3 | 316.3 | 416.4 | 268.0 |
| External click | 227.6 | 110.0 | 488.5 | 146.7 | 107.0 | 203.4 |
| Up click | 3,816.6 | 1,376.1 | 33,655.0 | 6,576.2 | 3,136.6 | 37,196.5 |
| All events | 56.6 | 25.8 | 140.6 | 66.5 | 43.7 | 109.0 |

literature was available through open access (and this fact was promoted among readers) health professionals may be more likely to pursue the greater educational advantage of the external links. Currently, there are very few links bearing an open access icon, and readers show no additional interest in viewing these links just because they are open. If readers start expecting that external links will be open (and there is some promotion of this fact), then the two

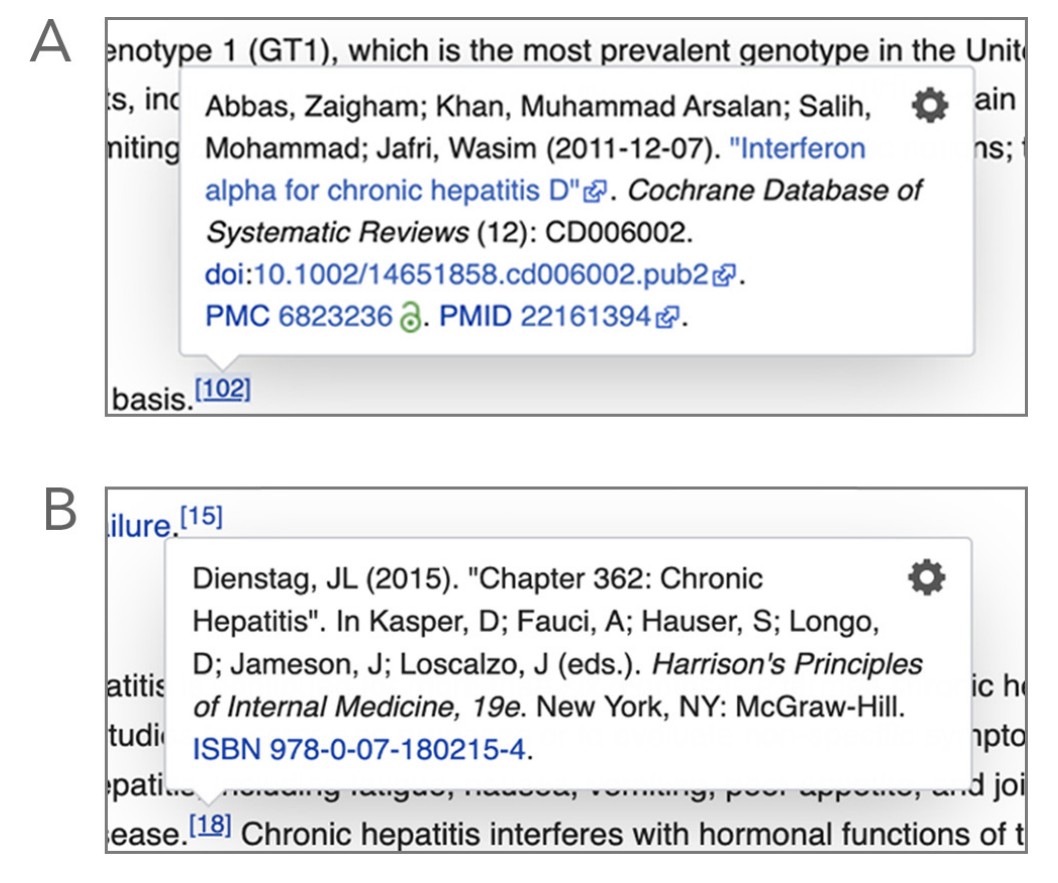

**Figure 3.** Example of information displayed when hovering over footnotes in Wikipedia. (A) An external link on the Hepatitis page, revealed by "hovering" over the footnote number, indicating that the source is a systematic review conducted by the Cochrane Collaboration (with a green icon PMC link signaling open access to the source). (B) Hovering over a different footnote number on the same page reveals a different external link for the 19th and relatively recent edition of a medical textbook from a leading publisher.

**Table 6.** Summary of data collected.

Summary statistics selected from the tables above, comparing WikiProject Medicine (WPM) pages and readers to the rest of Wikipedia (W) pages and readers. [a] The "4.4 more pageviews" reflects the ratio of WPM to W pages (0.6%) compared to the ratio of WPM to W pageviews/day (2.6%). [b] "WPM readers" and "W readers" refer to the behaviors of those reading a WPM and/or W page during the data collection period.

| | WPM pages | W pages | WPM pages, compared to other W pages,... |
|---|---|---|---|
| Page length (characters) | 13,085 | 7676 | ...are 70.5% longer by character count. |
| Characters/ external link (on pages with links) | 450.3 | 657.3 | ...possess a 31.5% greater external link density. |
| Pageviews/ day | 5,875,470.40 | 228,445,128.40 | ...receive 4.4 more pageviews per day.[a] |
| | **WPM readers** | **W readers** | **WPM readers, compared to rest of W readers,...**[b] |
| Time before engagement (sec) | 47.5 | 32.8 | ...take 44.8% longer before engaging in a link activity. |
| Pageviews/link engagement | 56.60 | 66.47 | ...engage 17.5% more often with links per pageview. |
| Pageviews/ hover | 120.5 | 203.5 | ...hover over links 68.9% more often per pageview. |
| Pageviews/ footnote click | 211.8 | 316.3 | ...click footnote numbers 49.3% more often per pageview. |
| Pageviews/ external click | 227.6 | 146.7 | ...click external links 35.5% less often per pageview. |

studies just cited suggest that the chances of them pursuing an item of interest will become more likely.

Although this study does not provide a means of determining why readers hover rather than click on external links, this is something that could be addressed using smaller-scale research strategies like reader questionnaires or think-aloud protocols. Alternatively, design experiments, which experiment with external link format and context, could be conducted involving, (a) WPM pages in which all external links are clearly indicated as open access, (b) tested readers are given prior training in how to learn from cited works, and (c) how different mobile interfaces impact the amount research studies are clicked on for further reading.

## Limitations

One of the limitations of this study is the way in which the count for "hovering" (defined as a reader's cursor lingering over a link for 1000 milliseconds or more, revealing a rollover label) captures both intentional and unintentional acts. So, while the reported number of hovers is of limited value, there is no reason to believe that the incidental hovers would differ between WPM and W readers. As well, with the pageview data,

various strategies were used to exclude pages visited by bots, but the limited effectiveness of these strategies is a known weakness in Wikipedia's infrastructure. It, again, suggests a tempering of the pageview counts but not the ratios between WPM and W in this regard. This is similarly the case with the exclusion from this study of interwiki links that reduced external link counts, as described in the methods. Lastly, this study focuses only on the English-language version of Wikipedia and therefore our findings are limited to this version of the encyclopedia.

## Conclusion

The study has identified a number of distinctions that set WPM pages apart from the rest of Wikipedia. These differences, such as page length and link density, reflect the quality of these pages as information sources. This finding was further supported by differences in reader behavior, such as time spent on page, and amount of external link engagement, particularly the rate readers hover or click on footnote numbers to examine the bibliographic information on the sources contained in the links. This suggests that WPM readers are more engaged in assessing the citation of the external link, which assists in validating and building trust in the

content that makes up the WPM pages, than by the opportunity to visit the sources as a gateway for further learning. Subsequent studies of readers engagement with Wikipedia's external links should investigate the role citations play in what readers learn about a topic and whether there is the potential, especially for readers in health professions, to increase the degree and depth of learning.

## Disclaimer

The views expressed in this article are those of the authors and do not necessarily reflect the official policy or position of the Uniformed Services University of the Health Sciences, the U.S. Department of Defense, or the U.S. Government.

## Acknowledgements

The authors wish to acknowledge the value of this opportunity to collaborate with the WMF research team, which began in 2018 with the posting of a research proposal to Wikimedia's Meta-Wiki. Members of the community, including members of WPM, were invited to provide feedback, supported by teleconferences and in-person meetings with the WMF research team on infrastructure, parameters of data use (e.g., best practices for accessing the data securely; the duration of data access), and the expectations of the Wikipedia community. As well, this collaboration extended to members of the Data Science Lab at École polytechnique fédérale de Lausanne (EPFL) who designed the CitationPageload schema, and with whom the authors developed the CitationUsage schema.

**Lauren A Maggio** is in the Department of Medicine, Uniformed Services University of the Health Sciences, Bethesda, United States

lauren.maggio@usuhs.edu

https://orcid.org/0000-0002-2997-6133

**Ryan M Steinberg** is in the Lane Medical Library, Stanford University School of Medicine, Stanford, United States

https://orcid.org/0000-0003-0101-4490

**Tiziano Piccardi** is in the École Polytechnique Fédérale de Lausanne, Lausanne, Switzerland

**John M Willinsky** is in the Graduate School of Education, Stanford University, Stanford, United States

https://orcid.org/0000-0001-6192-8687

*Author contributions:* Lauren A Maggio, Conceptualization, Formal analysis, Investigation, Methodology, Project administration; Ryan M Steinberg, Conceptualization, Data curation, Software, Formal analysis, Investigation, Methodology; Tiziano Piccardi, Data curation, Formal analysis, Investigation, Visualization, Methodology; John M Willinsky, Conceptualization, Formal analysis, Methodology

*Competing interests:* The authors declare that no competing interests exist.

## Funding

No external funding was received for this work.

## Decision letter and Author response

Decision letter https://doi.org/10.7554/eLife.52426.sa1
Author response https://doi.org/10.7554/eLife.52426.sa2

## Additional files

### Supplementary files

• Transparent reporting form

## Data availability

With the approval and support of the Wikimedia Foundation (WMF), we accessed the data presented in this study via the Wikimedia's Event Logging system and the production MediaWiki database. Due to WMF data policies intended to protect the privacy of readers and editors, all of the data collected and analyzed has had to remain within their protected data environment. Therefore, the data is not publicly available. While this data is not publicly available, data may be available upon request from the WMF research team. To gain access to the data, researchers should review the WMF Research Team's current procedures for data requests (https://www.mediawiki.org/wiki/wikimedia_research/research_and_data). The WMF Research Team will evaluate the request based on Wikimedia's data access criteria. The code utilized to collect and analyze the data, however, is organized and made publicly available in a collated series of Jupyter notebooks at: https://github.com/ryanmax/wiki-citation-usage/ (Steinberg et al., 2019; copy archived at https://github.com/elifesciences-publications/wiki-citation-usage).

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
