## [Decision Letter]

Thank you for submitting your article "Reader engagement with Wikipedia's medical content" for consideration by *eLife*. Your article has been reviewed by three peer reviewers, and the evaluation has been overseen by two members of the *eLife* Features team (Julia Deathridge and Peter Rodgers). The following individuals involved in review of your submission have agreed to reveal their identity: James Heilman (Reviewer #1); Besnik Fetahu (Reviewer #2); Shani Evenstein (Reviewer #3).

The reviewers have discussed the reviews with one another and we have drafted this decision to help you prepare a revised submission. We hope you will be able to submit the revised version within two months.

Summary:

This work enriches our understanding of how readers of Wikipedia's medical pages engage with external links compared to readers of standard Wikipedia pages. However, there are a few major comments that need to be addressed.

Major comments to address:

a) The results reveal that in general the readers of the standard Wikipedia pages are more prone to actually clicking the external links. This is interesting given that one would think that the motivation for visiting such pages may reflect more personal reasons, i.e. checking on symptoms of a disease. Yet, the non-WPM pages seem to have better engagement with the external links. One possibility for this could be that the page visits are not normally distributed across all the pages. You have highly important pages in Wikipedia that gather a lot of attention, thus, their external links are more likely to be visited. I assume that normalizing the Wikipedia articles according to their page visits might paint a different picture here with respect to the page visit.

b) In the second research question, the motivation behind visiting such external links is postulated. While personally I agree with the points in the discussion as to why this may happen, there is no actual evidence for this in the current study. The authors should include a sentence or two discussing this limitation and what further work could be done to validate their hypotheses, such as carrying out a questionnaire with a diverse set of WPM readers that investigates their reasoning for clicking/not-clicking on external links.

c) As the article aims to review WPM content as a gateway to further learning and deeper understanding of a given topic, especially as it is used by both students, health professionals and the general public, one thing that was missing from this effort was an acknowledgement of the importance of both internal links and media files as a means of enhancing understanding, not only that of references and footnotes. Even if this research effort chooses to focus only on external links and footnotes, then an explanation is expected as to why specifically this focus. Especially in a world where imagery is so dominant, and videos have become a major means of education, further research is needed regarding how readers engage with media content on WPM articles, compared with the rest of Wikipedia. If the research team has any data related to this, not only to external links and footnotes, it could be of great contribution to mention it as well.

d) While research till 2016 (Heilman & West) showed that Wikipedia was the most popular site for medical content online, more than sites like WHO, NIH etc, WikiProject Medicine Foundation has shown that this has changed in the last couple of years. It seems that Wikipedia's medical content is no longer the most visited site in the world for medical content and is "losing" to sites that encourage users to ask different questions on the same topic and have multiple answer pages on that same topic. It would be of interest to the public if the research team considers this and discusses the results in this light. It might offer additional conclusions to the ones already presented.

---

## [Author Response]

[We repeat the reviewers’ points here in italic, and include our replies point by point, as well as a description of the changes made, in plain text]

Major comments to address:a) The results reveal that in general the readers of the standard Wikipedia pages are more prone to actually clicking the external links. This is interesting given that one would think that the motivation for visiting such pages may reflect more personal reasons, i.e. checking on symptoms of a disease, yet, the non-WPM pages seem to have better engagement with the external links. One possibility for this could be that the page visits are not normally distributed across all the pages. You have highly important pages in Wikipedia that gather a lot of attention, thus, their external links are more likely to be visited. I assume that normalizing the Wikipedia articles according to their page visits might paint a different picture here with respect to the page visit.

We appreciate this question on our finding that WikiProject Medicine (WPM) readers clicked on external links less often per view than Wikipedia (W) readers. We investigated ways of normalizing for pageviews and page-length, as two factors that might be associated with this result. The most common approach would be to use matching methods, involving creating matched samples or datasets for, in this case, matching WPM and W for pageviews and for page length. This creates, in effect, random samples for these two factors which then allows one to determine the effect of an intervention, or in this case the extent to which a factor is associated with an external link behavior. But since we set out to determine if and how WPM pages and readers differ from the rest of W pages and readers, our goal has been to describe the pattern of the differences as a whole, rather than ask what would happen if pageviews or page lengths were the same for WPM and W. This helped us realize that the focus on the comprehensive pattern was not previously clear enough, and we have now, thanks to this reviewer’s comment, brought it to the fore in revisions to the research questions and discussion. The principal change is to have introduced a summary table (Table 6) in the discussion that presents the three key factors that distinguish WPM pages from the rest of W, and the five key factors that distinguish WPM readers from W readers. As a result of structuring the results this way, we believe it is easier to grasp consistencies that are to be expected with the reading behaviors of health sciences students and professionals (whose use of WPM has been well established, if not the proportion of WPM readers that they make up). We think that this points to the sort of future research helpfully suggested by a reviewer involving questioning readers either directly about their choices or indirectly through think-aloud protocols. All of that said, if the reviewers still feel that matching methods should be applied, we would, of course, be willing to conduct such an analysis. To reflect the above changes, we have also modified the abstract.

b) In the second research question, the motivation behind visiting such external links is postulated. While personally I agree with the points in the discussion as to why this may happen, there is no actual evidence for this in the current study. The authors should include a sentence or two discussing this limitation and what further work could be done to validate their hypotheses, such as carrying out a questionnaire with a diverse set of WPM readers that investigates their reasoning for clicking/not-clicking on external links.

We interpreted this important point as partly a valuable critique of our second research question. We have reframed the research questions in the introduction so that they follow the pattern of the study, dealing with differences between WPM and W pages and readers. This had led us to extend the analysis of the greater hovering and footnote-number clicking in the Discussion, in which WPM readers must be gaining additional reassurances from how the title and publication venue of the references can provide scientific characteristics of the study, using the “ideal sources” of systematic reviews and medical textbooks as examples. That said, we fully embrace and now include the reviewer’s point about follow-up qualitative studies among a diversity of users to determine what influences their choices around engagement with external links.

c) As the article aims to review WPM content as a gateway to further learning and deeper understanding of a given topic, especially as it is used by both students, health professionals and the general public, one thing that was missing from this effort - an acknowledgement of the importance of both internal links and media files as a means of enhancing understanding, not only that of references and footnotes. Even if this research effort chooses to focus only on external links and footnotes, then an explanation is expected as to why specifically this focus. Especially in a world where imagery is so dominant, and videos have become a major means of education, further research is needed regarding how readers engage with media content on WPM articles, compared with the rest of Wikipedia. If the research team has any data related to this, not only to external links and footnotes, it could be of great contribution to mention it as well.

We did fail to properly acknowledge the contribution of internal links for reader learning. And while our focus remains on the association between Wikipedia and the body of external research, that has now been made clear (in the Introduction), in the context of how richly cross-referenced Wikipedia is and how much of that lateral reading can contribute to the informational and educational quality of WPM pages.

d) While research till 2016 (Heilman & West) showed that Wikipedia was the most popular site for medical content online, more than sites like WHO, NIH etc, WikiProject Medicine Foundation has shown that this has changed in the last couple of years. It seems that Wikipedia's medical content is no longer the most visited site in the world for medical content and is "losing" to sites that encourage users to ask different questions on the same topic and have multiple answer pages on that same topic. It would be of interest to the public if the research team considers this and discusses the results in this light. It might offer additional conclusions to the ones already presented.

This comment raises interesting points about changes to the questions readers may be asking and how that is affecting the popularity of WPM pages. After some discussion among us, as well as searches for relevant literature on it, we wish to respectfully suggest that this comment is out of scope for this paper, given our focus on reader engagement with external links on WPM pages. Based on recent studies on medical and health provider use of Wikipedia (Scaffidi, 2017), we see the relevance of this site for health information is still substantial, and a particularly relevant source for the initial and continuing education of health professionals. We appreciate the reviewers’ encouragement to consider broader issues around such topics as reader interests and questions.